



# Transport of Canadian forest fire smoke over the UK as observed by lidar

Geraint Vaughan[1,2], Adam P. Draude[2], Hugo M. A. Ricketts[1,2], David. M. Schultz[2], Mariana Adam[3,5], Jacqueline Sugier[3], and David P. Wareing[4]

[1]National Centre for Atmospheric Science, University of Manchester, UK
[2]School of Earth and Environmental Sciences, The University of Manchester, UK
[3]Met Office, Exeter, UK
[4]Aberystwyth University, UK
[5]National Institute of R&D for Optoelectronics, Magurele, Romania

*Correspondence to:* G. Vaughan
geraint.vaughan@manchester.ac.uk

**Abstract.** Layers of aerosol at heights between 2 and 11 km were observed with Raman lidars in the UK between 23 and 31 May 2016. A network of such lidars, supported by ceilometer observations, is used to map the extent of the aerosol and its optical properties. Spaceborne lidar profiles show that the aerosol originated from forest fires over Western Canada around 17 May, and indeed the aerosol properties – weak depolarisation and a lidar ratio at 355 nm in the range 35–65 sr – were consistent with long-range transport of forest fire smoke. The event was unusual in its persistence – the smoke plume was drawn into an atmospheric block that kept it above North-west Europe for nine days. Lidar observations show how the smoke layers became optically thinner during this period, but the lidar ratio and aerosol depolarisation showed little change.

## 1 Introduction

Forest fires occur every summer over the boreal forest of the Northern Hemisphere (Weber & Stocks, 1998; Wooster & Zhang, 2004). The smoke from these fires can be lifted to great heights by deep convection – indeed, the fires can amplify the storms leading to so-called pyroconvection (Fromm, 2005). Once deposited in the free troposphere or stratosphere, forest-fire smoke can travel great distances, e.g. from North America to Europe (Forster et al., 2001; Müller et al., 2005) or even around the globe (Damoah et al., 2004). In this paper we discuss such a transport event that occurred in May 2016, when smoke from intense fires in Western Canada reached Europe and was observed by the UK lidar network. The fires in this case caused headlines around the globe due to the destruction of the Canadian town Fort McMurray (56.72°N, 111.38°W) in north-east Alberta (https://en.wikipedia.org/wiki/2016_Fort_McMurray_Wildfire).

Raman lidars have been used extensively to study long-range smoke transport, measuring optical and microphysical properties and calculating the age and origin of the smoke (e.g. Müller et al., 2005, 2007; Ansmann et al., 2009; Amiridis et al., 2009; Giannakaki et al., 2010; Pereira et al., 2014; Veselovskii et al., 2015). These studies have found smoke particles to be weakly depolarising (i.e. slightly non-spherical) and to have effective radii of less than 1 $\mu$m. The extinction to backscatter ratio (lidar ratio, LR) of aged smoke depends on its age and origin, and is similar to that from other pollution aerosols (Müller





et al., 2007). We show here how a dense network of lidars and ceilometers tracked the evolution of a smoke episode from 23 to 31 May 2016 as an atmospheric block trapped the air over Western Europe. Spaceborne lidar data from the CALIOP and CATS instruments, supported by SEVERI images from Meteosat-10, enables the smoke to be tracked back unambiguously to the fires over Alberta on 17 May.

## 2 Instrumentation

The aerosol cloud was measured by a number of lidars around the UK (Fig. 1):

- the Raman lidar at the Natural Environment Research Council (NERC) Mesosphere–Stratosphere–Troposphere (MST) Radar facility at Capel Dewi (52°25'N, 4°0'W), Wales

- the Met Office Raman lidars located at Camborne (50°12'N, 5°17'W), East Malling (51°17'N, 0°26'E), Exeter (50°43'N, 3°28'W), Loftus (54°31'N, 0°53'W) and Watnall (53°0'N, 1°15'W)

- twelve Lufft CHM 15k ceilometers operated as part of the Met Office's UK ceilometer network

A brief description of these facilities is now given.

### 2.1 Capel Dewi Raman Lidar

The Capel Dewi Raman lidar is a biaxial ultraviolet lidar based on that used for the EARLINET project in 1999–2002 (Wandinger et al., 2004). Since then, it has been updated and now contains a Continuum 8030 Nd-YAG laser emitting pulses at 354.7 nm at 30 Hz with pulse energy 300 mJ. A tenfold beam-expanding telescope directs the light vertically into the atmosphere. The receiver is based on a 1 m diameter mirror used in a Nasmyth-Cassegrain configuration, which directs the backscattered radiation through a collimator on to a dichroic beamsplitter (Fig. 2). This beamsplitter reflects and transmits radiation with wavelength greater or less than 397 nm, respectively, into the receiver channels. Interference filters centred around 387 and 408 nm isolate Raman scattering from nitrogen and water vapour respectively (Table 1). A third channel measures elastic backscattered radiation reflected from the other two filters. The receiver is mainly sensitive to the polarisation component parallel to the laser, which reduces background noise in the Raman channels but does not permit measurements of the aerosol backscatter when there is a significant cross-polarised component.

The lidar is designed for free tropospheric measurements and so the receiver field-of-view does not fully overlap the laser beam below 2 km. Measurements below 2 km are therefore not used here.

The signals are measured using EMI 9124 photomultipliers and a photon-counting electronics system (ORTEC PCI-MCS) with range resolution 15 m (100 ns time bins). A dead time of 10 ns is applied in the counting system, and corrected according to the non-paralysable equation:

$$S = \frac{S_0}{1 - \tau * S_0} \tag{1}$$

where $S_0$ is the measured count rate, $S$ is the corrected count rate and $\tau$ the dead time.



| Channel | Centre Wavelength nm | Bandwidth FWHM nm | Measured Wavelength $\lambda_m$, nm | Transmission at $\lambda_m$, % |
|---|---|---|---|---|
| $H_2O$ | 408 | 1.7 | 407.5 | 22 |
| $N_2$ | 387 | 2.6 | 386.7 | 22 |
| Elastic | 355 | 5.0 | 354.7 | 60 |

**Table 1.** Interference filter characteristics for Capel Dewi lidar (FWHM is full width half maximum). Blocking on the Raman filters at 354.7 nm is $10^{-8}$. Neutral density filters are used in the elastic channel to avoid saturating the signal at low altitude.

Although elastic measurements can be made in daytime, the Raman signals are too noisy in daytime and are only collected at night. The system is normally operated alongside a second lidar which measures both polarisation components of the elastic signal, but this system was inoperative during the period of interest here.

Measurements were made with the Capel Dewi lidar on the nights of 23–24, 24–25, 26–27, 29–30 and 30–31 May; persistent low cloud cover precluded measurements during the other nights.

## 2.2 Met Office Raman Lidar

The Met Office has established an operational network of Raymetrics Raman and depolarisation lidars around the UK. These are used alongside ceilometers, airborne in situ and remote–sensing observations, satellite retrievals and dispersion model output for volcanic ash monitoring by the London Volcanic Ash Advisory Centre (VAAC) (Marenco et al., 2016). Like the Capel Dewi lidar, these Raman lidars use the third harmonic of a Nd-YAG laser (Quantel CFR200) at 355 nm. The lasers provide a pulse energy of 50 mJ at 20 Hz, directed vertically via a beam expander to assure eye-safety. The receiving telescopes are 30 cm in diameter with complete overlap achieved around 250 m altitude (Adam et al., 2016). The range resolution of the receiver is 15 m. Unlike the Capel Dewi lidar, these lidars measure both parallel- and perpendicularly-polarised returns from the atmosphere and therefore provide measurements of aerosol backscatter, depolarisation and lidar ratio as well as optical depth. The method used to calibrate the depolarisation, along with a layout of the receiving unit, is described by Buxmann et al. (2017).

The lidar has both analogue and photon-counting detection channels, although in this paper only the photon-counting measurements are used. A deadtime correction as above with $\tau$=3.8 ns is applied to these data. As with the Capel Dewi lidar, Raman measurements can only be used at night, which for the episode considered here (at the end of May in the UK) meant 2100–0300 UTC. The lidars were operated in one of two modes: intermittently for periods of 1 hour every 3 hours, or continuously, but not all of them were operational at any given time. Table 2 summarises the availability of data from the Met Office Raman lidars.





| Date | 23 | 24 | 25 | 26 | 27 | 28 | 29 | 30 | 31 |
|------|------|------|------|------|------|------|------|------|------|
| Camborne | IC10 | C | C | C | C | C | C | C | C |
| E. Malling | | | I | IC10 | C | C | C | C | I |
| Exeter | | | I | IC12 | C | C | C | C | I |
| Loftus | IC15 | C | | C11 | 09C | | | | |
| Watnall | | | I | IC10 | C | C | C | C | C |

**Table 2.** Data coverage from Met Office lidars. I denotes intermittent coverage (1 hour in 3), C denotes continuous coverage. IC10 denotes intermittent coverage up to 1000 UTC then continuous thereafter. C11 denotes no coverage up to 1100 UTC then continuous thereafter; 09C denotes continuous coverage up to 0900 UTC and none thereafter.

## 2.3 Met Office Ceilometers

The Met Office operate an extensive network of ceilometers around the UK, of various types. The 12 Lufft CHM 15k ceilometers are of particular interest to this study, because of their greater sensitivity to thin aerosol layers with low backscatter. These ceilometers emit infrared radiation (1064 nm) and use photon-counting detectors. While they cannot provide the quantitative detail of the Raman lidars, they operate continuously and can provide more complete coverage in space and time of the free–tropospheric aerosol.

## 3 Retrieval of aerosol optical depth and lidar ratio

The power received by a lidar, $P(z)$, obeys the lidar equation (Wandinger, 2005), which for elastic ($P_e$) and Raman ($P_R$) scattering takes the form:

$$P_e(z) \propto [\beta_{aer}(z) + \beta_{ray}(z)] exp[-2 \int_0^z (\sigma_{ray} n(z) + \alpha(z)) dz] \qquad (2)$$

and

$$P_R(z) \propto \beta_{ram}(z) exp[- \int_0^z ((\sigma_{ram} + \sigma_{ray}) n(z) + 2\alpha(z)) dz] \qquad (3)$$

respectively. Here, $\beta_{aer}$, $\beta_{ray}$ and $\beta_{ram}$ are the backscatter coefficients for aerosol, elastic molecular (Rayleigh) and Raman scattering respectively, n(z) is the number density of air molecules, z is the height above the lidar, and $\sigma_{ram}$ and $\sigma_{ray}$ are the scattering cross-sections for Raman and Rayleigh scattering by air molecules, which are taken to be $1.929 \times 10^{-30}$ m$^2$ and $2.76 \times 10^{-30}$ m$^2$ respectively (Bates, 1984). The extinction coefficient of aerosol, $\alpha$, is assumed to be the same at the elastic and Raman wavelengths.

Retrieval of the aerosol optical depth uses the N$_2$ Raman signals and a nearby radiosonde profile. From the latter, a synthetic molecular-only scattering profile may be constructed, and fitted to the measured profile in a region of the atmosphere free from



aerosol, here taken to be 13–16 km. The ratio $Ram(z)$ of the measured to the synthetic signals then leads to curves such as that shown in blue in Fig. 3. In principle, this ratio can be inverted to give a profile of $\alpha(z)$, but this approach tends to lead to large random errors. We take advantage here of the fact that in the episode under discussion the aerosol was distributed in very distinct layers, so a method was devised to calculate only layer-average or layer-total quantities. Figure 3 also shows $R(z)$, the

ratio of the elastic channel to the Raman channel, normalised to 1 between 13 and 16 km. Aerosols show up in this curve as departures from 1 (the molecular background), clearly showing the layered structure. (A small correction has been applied to $R(z)$ to account for the difference between $\sigma_{ram}$ and $\sigma_{ray}$, using the radiosonde profile). Such structure was observed at all sites during the course of this event. We therefore calculate the integrated aerosol optical depth (AOD) across each layer:

$$AOD = -\frac{1}{2}ln(\overline{Ram(above)}/\overline{Ram(below)}) \qquad (4)$$

where the overbars indicate that $Ram(z)$ has been averaged in the aerosol-free regions above and below each layer. Each lidar profile used here was examined separately to determine the layer altitudes and the width of the aerosol-free regions, which were chosen as far as possible to be at least 1 km deep.

As the photon-counting signals can be assumed to follow Poisson statistics, the precision error in AOD is readily calculated from the number of photon counts in the regions above and below each layer. A further source of error comes from the choice

of radiosonde profile used to normalise $P_R(z)$. For stations like Camborne and Watnall where co-located radiosondes were released, this error is small, but for the other stations it is not negligible. As an example, Fig. 4 shows the same lidar data as Fig. 3, but using a radiosonde from Camborne rather than Castor Bay (Fig. 1). The difference in AOD for the five layers is greater than the statistical uncertainty, showing that this source of error is important for free tropospheric aerosol measurements by Raman lidar.

Retrieval of the integrated aerosol backscatter, IAB, and hence the mean lidar ratio LR (LR = AOD/IAB) for each layer requires measurement of both polarisation components, and this was only possible for the Met Office Raymetrics lidars. Signals from the two elastic channels were added (after calibration of the cross-polarised channel as described by Buxmann et al. (2017)), and used to generate an $R(z)$ profile as before. The integral of B[$R(z)$ - 1]n(z) across each layer then gave IAB (where B = $3.31 \times 10^{-31}$ m$^2$ sr$^{-1}$ is the molecular differential backscatter cross-section, after Bates, 1984). Errors in IAB

come from the Poisson statistics in both the signals in the layer and the background noise subtracted from the measured lidar signals, which are treated differently under the integral (variances being added for the signals and standard deviations for the background). Finally, the two errors are added in quadrature to give the error in IAB, and LR calculated for each layer in the usual way.

## 4   Results

To gain an appreciation of the extent and persistence of the aerosol, we first examine the ceilometer measurements since continuous coverage was available from all of them throughout the period, with aerosol measurements limited only by the presence of cloud.





Thin layers of free-tropospheric aerosol began to appear over the UK during 22 May 2016, and persisted intermittently until the end of the month. As an example, Fig. 5 shows the variation of backscatter signal with height and time of the Lufft ceilometers at Lerwick (60°8'N, 1°11'W), Dishforth (54°7'N, 1°24'W) and Camborne (50°12'N, 5°17'W) on 23 May. Thin layers of enhanced backscatter due to aerosols are shown throughout the day at Camborne, and from 1000 UTC onwards at the

5 other two stations. Similar patterns are seen in the layers at all three stations, suggesting that the aerosol layer was widespread over the UK during the second half of 23 May. To demonstrate this further, Fig. 6 summarises the ceilometer observations during 23 and 24 May. Four categories are shown: those with aerosol between 4 and 8 km (blue), those with aerosol between 2 and 4 km but no higher (green), those with no or only a trace of aerosol (red) and those where cloud cover precluded observations (yellow). High-altitude aerosol (>4 km) was indeed widespread across the UK on both days, with a suggestion

of a clearance in the far north on the 24th.

To examine the duration of the event, the total number of ceilometers which observed definite aerosol layers, trace amounts or no aerosol, or were restricted by cloud cover, was plotted for each day from 22 to 27 May (Fig. 7). This plot shows that the event seems to have peaked in terms of coverage on 24 May, although the increasing cloud cover thereafter means that some aerosol is likely to have been missed. None of the ceilometers detected aerosol after the 27th, although continuing low cloud

cover restricted observations to around half the stations until clear skies returned on 5 June.

The ceilometers only provide consistent measurements up to 8 km, and their infra-red wavelength means they can only give qualitative information on the presence or not of aerosols. To extend the measurements to the tropopause and obtain quantitative information about the aerosol, we now turn to the Raman lidars. A qualitative inspection of the Met Office elastic channels (parallel and perpendicular polarisation) showed that there was extensive aerosol between 8 km and the tropopause which was

20 not captured by the ceilometers. This was observed at Camborne during 24–28 May, East Malling from 25–27 May, Exeter from 25–31 May and Watnall from 26–31 May. The Capel Dewi lidar also observed aerosol between 8 and 12 km up to the end of May. This shows that the event persisted from 22 to 31 May with aerosol layers found at all altitudes in the troposphere. We now concentrate on the night-time measurements from the Raman lidars to obtain quantitative information on these layers.

A similar analysis to that described in section 3 was conducted for all the continuous night-time data collected by the Raman

lidars between 23 and 31 May, after eliminating periods affected by cloud. Figure 8 shows the total aerosol optical depth above 2 km measured by the Raman lidars during this period, aggregated into whole-night averages. The greater sensitivity of these lidars means that aerosol was measured up to the night of 30–31 May at Capel Dewi with traces evident up to the same time at some of the other stations. However, the coverage is patchy due to the combination of intermittent sampling and low cloud cover.

The highest AOD measured was that at Capel Dewi on the night of 23–24 May, from the profile shown in Fig. 4. Although the value is sensitive to the choice of radiosonde profile, the total AOD of ~0.13 above 2 km clearly results from aerosol observed throughout the free troposphere. All other AOD measurements at all the stations were below 0.1, with the values decreasing with time to below 0.05 after the 28th.

The evolution of lidar ratio as a function of optical depth is shown for the Met Office lidars in Figs. 9 and 10, for aerosol

layers above and below 7 km respectively. Here, hourly average data are presented, as there could on occasion be considerable





variability in AOD during a night; an example is 27–28 May at Camborne where the total AOD varied between 0.02 and 0.09 over the 5-hour measurement period. With the exception of a few outliers, the lidar ratios generally fall between 35 and 65 sr, consistent with the lidar ratios of 21–67 sr at 355 nm for Canadian and Siberian smoke reported by Müller et al. (2005), and with the values of $46 \pm 13$ sr found in the 10-year study of forest-fire smoke by Müller et al. (2007) using EARLINET data.

There seems to be a greater spread of lidar ratios in the upper-tropospheric layers than in the mid-troposphere, where the spread is more like 40–60 sr. Note that the abscissa scales in Figs. 9 and 10 are different: most of the aerosol measured in this dataset was found in the upper troposphere, where AOD values generally extended to 0.07, compared to 0.035 in the mid-troposphere.

The volume depolarisation ratio for the lidar profiles containing aerosol layers was a few percent at 355 nm, consistent with optically thin layers embedded within strong molecular backscatter. However, when converted to aerosol depolarisation ratios,
$\delta_a$, a remarkably consistent picture emerged across the different stations. Below 7 km, the value of $\delta_a$ was in the range 0.04–0.06, whereas above 7 km it was generally close to 0.20: for all four of the nights at Camborne and two of those at Watnall, $\delta_a$ lay in the range 0.18–0.21. Lower values were measured above 7 km at Exeter and East Malling on 25–26 May (0.15) and a high value at Watnall on 26–27 May (0.31) but the clear altitude difference remains.

A higher value of $\delta_a$ means the aerosol particles are more depolarising, which suggests more irregular solid shapes. As all
the measurements here were made using the same laser wavelength, we cannot infer anything about particle size from the data, but the greater $\delta_a$ at higher altitudes is consistent with some of the smoke particles having acquired an ice coating at the colder temperatures near the tropopause - consistent with the ice-nucleating potential of smoke particles identified by Tan et al. (2014).

## 5  Origin of aerosols

Having observed the presence of aerosol layers over the UK, three questions need to be answered. Where did they come from? How old are they? What height(s) was the aerosol injected into the atmosphere?

Copernicus Atmospheric Monitoring Service (CAMS) daily fire products[1] showed extensive forest fires in Canada during May 2016, especially in Saskatchewan and Alberta (e.g. Fig. 11). Given the prevailing westerly winds in midlatitudes, and the presence of deep convection over Canada to lift the smoke, these provide the most likely source for the aerosol found over the
UK. This hypothesis will now be examined using meteorological charts, trajectory calculations and satellite observations.

The air flow in the free troposphere impinging the UK on 20 May (Fig. 12a) was zonal, with rapid flow across the Atlantic around a trough at 54°N, 32°W. This provided a route to bring smoke aerosol across from Canada. After this, the pattern became more complicated. The trough moved steadily eastward and deepened, with its axis along the Irish Sea by 1200 UTC on the 22nd (Fig. 12b). At the same time, a deepening depression east of Newfoundland resulted in a second trough near 45°N,
40°W. These two troughs, and the ridge in between, set up an omega block during the 23rd which resulted in a split jet stream, with one branch heading north over Iceland towards the Norwegian Sea, and the other heading south across the Mediterranean (Fig. 12c). (An omega block is characterised by an upper–level ridge or anticyclone flanked by two cut-off lows or troughs,

---

[1] available from http://macc.copernicus-atmosphere.eu/d/services/gac/nrt/fire_radiative_power





to the south-west and south-east.) As the block moved and distorted, the UK lay first under the eastern trough (0600 UTC on the 22nd to 1200 UTC on the 23rd), then the anticyclone (1800 UTC on the 23rd to 1200 UTC on 24th, Fig. 12c), the western cut-off low (1800 UTC on the 24th to 0000 UTC on the 29th, Fig. 12d) and finally a broad area of almost no flow which persisted until a second, weaker omega block was established on the 30th as another depression developed in the western

Atlantic and moved eastwards (not shown). From 23–31 May therefore the flow over the UK was slack and variable, which meant that smoke transported in the zonal jet up to the 22nd was able to remain in the vicinity of the UK.

## 5.1   Air parcel trajectories

We now examine air parcel trajectories for evidence that the aerosol-laden air crossed the Atlantic from Canada. To be useful for this purpose, trajectories need to be non-dispersive – i.e. trajectories from nearby starting points need to follow a similar

path. Unfortunately, this did not prove to be the case for most of this event, precluding any meaningful conclusion on air-mass origin. We concentrate therefore on the period leading up to the start of the event when coherent sets of trajectories were found.

Trajectories were calculated using NOAA's HYSPLIT trajectory model (Draxler & Hess, 1998; Stein et al., 2015), both backward in time from the locations of the lidars and forward in time from locations in Western Canada. A matrix of 9 starting points was defined, spaced by 0.5° in latitude and longitude; low dispersion of these 9 trajectories is required if the calculations

are to be considered reliable.

As an example, Fig. 13 (left panel) shows backwards trajectories from 11 km above Capel Dewi at 0000 UTC on 24 May, corresponding to the lidar profile in Fig. 3. Also shown are the forward trajectories at the same height from above Fort McMurray (56.72°N, 111.38°W) at 2000 UTC on 17 May, when cumulonimbi occurred over the fires. In both cases, the trajectories are sufficiently consistent to suggest that air passing over the UK on 23–24 May originated over the fire region

of Western Canada on the 17th. Following up on this, Fig. 14 shows the result of running the HYSPLIT on-line dispersion model with particles initiated in a column between 3 and 10 km over Fort McMurray at 2000 UTC on 17 May. This shows how any smoke particles lifted by convection would be transported eastward towards the Atlantic by 0800 UTC on the 21st (the HYSPLIT on-line dispersion model only runs for 84 hours).

However, examples like this proved rare. At other heights on 24 May, the back-trajectories from Capel Dewi were too

dispersive to reveal an air mass origin – by then the block was well set up with slack, incoherent flow. We therefore turn to satellite observations for evidence that the smoke crossed the Atlantic.

## 5.2   Satellite data

Several sources of data were used to track the smoke plume from Canada to the UK:

  1. The CALIOP lidar on board the CALIPSO satellite measures backscatter at 532 and 1064 nm, and depolarisation at

1064 nm (Winker et al., 2009). Plots of the data are available from http://www-calipso.larc.nasa.gov: these include backscatter, volume depolarisation and various other derived products. Plots of the version 4.10 products are used in this study. As well as the aerosol classification provided by NASA, smoke is expected to display low depolarisation: Pereira



et al. (2014) measured depolarisation values of 5% or lower for forest-fire smoke. This is consistent with the volume depolarisation of <6% measured by the Raman lidars over the UK.

2. The CATS lidar aboard the International Space Station (ISS) is similar to CALIOP, with channels at 532 and 1064 nm. Data plots at http://cats.gsfc.nasa.gov/data/browse/ were examined for indications of smoke-like aerosol. Smoke was identified as aerosol which appears in the total backscatter but not in the perpendicular backscatter, again consistent with the weak volume depolarisation.

3. The Ozone Mapping Profiler Suite (OMPS) instrument on the SUOMI-NPP satellite measures an Aerosol Index AI (Hsu et al., 1999), defined as the difference in the fraction of radiances, I, received at 331 nm and 360 nm to those calculated for a pure molecular atmosphere:

$$AI = -100(log_{10}[(\frac{I_{331}}{I_{360}}{}_{meas}] - log_{10}[(\frac{I_{331}}{I_{360}}{}_{calc}]) \tag{5}$$

The AI is defined such that UV-absorbing aerosols have positive values proportional to AOD (Hsu et al., 1999). For the purpose of this article, the AI is used as a measure of the presence of aerosol and an approximate guide to the amount of it.

4. The SEVIRI instrument aboard the Meteosat-10 satellite provides a geostationary view of the Earth. Natural Colour RBG images provided by EUMETSAT were examined every 15 minutes from 0300 UTC 22 May to 1945 UTC 24 May. In these images, smoke appeared as a faint blue-grey colour and was most distinct just after dawn and just before dusk, when the scattering of sunlight towards the satellite from the small smoke particles was more prevalent.

The eastward transport of aerosol across North America and over the Atlantic Ocean in Figs. 13 and 14 is shown by the OMPS-AI measurements (Fig. 15). The broad shape of the smoke plume heading eastward from Alberta is consistent with the HYSPLIT trajectories shown in Fig. 13, and shows aerosol reaching the western Atlantic on the 20th. Thereafter, the smoke tends to disperse and progress eastward towards Europe, with a strip of elevated aerosol index lying west of Ireland by 0500 UTC on 22 May.

The passage of smoke eastward can also be followed in night-time CALIOP data. Figure 16 shows backscatter, depolarisation and aerosol characterization between 0934 and 0939 UTC on 18 May 2016 from an orbit passing near 104°W, around 400 km east of Fort McMurray. CALIOP identified smoke between 3 and 8 km at the northernmost end of the orbital section, where aerosol depolarisation was <10%. On the previous orbit, at ∼0800 UTC roughly along 80°W between 53.5°N and 58°N, smoke was also present, again between 3 and 8 km, but none of the orbits further east observed smoke on this day. By the 20th, an orbit along ∼57°W around 0605 UTC measured mixed smoke and polluted aerosol from 49 to 56° N between 5 and 11 km (Fig. 17). Extensive smoke was also observed between 4 and 11 km on the 21st north of 49°N along ∼40°W and on the 22nd north of 50°N along ∼30°W (not shown). By the early hours of the 23rd, smoke layers had reached the vicinity of the UK (Fig. 18), again consistent with the trajectories.

The presence of smoke across the Atlantic Ocean is best shown by the total and perpendicular backscatter measurements by the CATS lidar in the early hours of 22 May (Fig. 19). Optically thin aerosol is identified as the light blue layers in the total





backscatter plot, and further classified as smoke by the lack of such layers in the perpendicularly polarised signal. Figure 19 shows that by 22 May the smoke plume extended from 55°W to 15°W and was present from the top of the boundary layer to above 10 km.

The EUMETSAT Natural Colour RGB analysis of SEVIRI data shows the arrival of smoke over the UK (Fig. 20). At 1830 UTC on 22 May, a ribbon of smoke extended from northern Spain to Iceland, passing west of Ireland. By 0445 UTC on the 23rd, this ribbon lay along the Irish Sea, just passing over Camborne at the western tip of Cornwall (consistent with the ceilometer evidence). By 1945 UTC on the 23rd, smoke covered most of the west of the UK and Ireland, consistent with Fig. 6a, and shows a similar pattern 24 hours later.

The combination of CALIPSO and CATS space-borne lidar, together with OMPS and SEVERI, therefore shows that a plume of smoke was drawn from Canada between 17 and 20 May which was transported eastward by the zonal flow during this period. Later, as the flow became blocked, this smoke was becalmed over the UK, and was observed by the UK lidar network. We have therefore shown that the aerosol observed by the Raman lidar at Capel Dewi and Met Office lidar network on 23 and 24 May had a consistent origin from the Canadian forest fires in western and central Canada around 17 May. This gives the smoke a 6–8-day transport time.

## 6 Conclusions

This study has presented observations of free-tropospheric aerosols by ceilometers and Raman lidars over the UK from 23–31 May 2016, and examined the origin of the aerosol. The principal conclusions are as follows.

- Ceilometer measurements showed that much of the United Kingdom was covered by free-tropospheric smoke layers on 23 and 24 May.

- Raman lidar observations showed that the smoke was found throughout the troposphere, but with the greatest optical depth above 7 km.

- The maximum optical depth measured was ∼0.15 with most values between 0.1 and 0.05: these values diminished with time through the event.

- The properties of the aerosol as determined from Raman lidar were consistent with those of smoke from forest fires: low volume depolarisation (<6% and a lidar ratio in the range 35–65 sr).

- The smoke lingered over western Europe for nine days due to an atmospheric block which prevented eastward advection.

- Although trajectory calculations proved indecisive for identifying the origin of the smoke, analysis of satellite lidar observations showed how the plume was drawn out over the Atlantic during 17–21 May before becoming becalmed by the block that developed on the 22nd.



The study shows the value of combining different kinds of lidars in following the evolution of long-range smoke transport events.

*Acknowledgements.* We thank the following individuals and organizations for providing access to data and imagery: Colin Seftor (NASA) for producing the OMPS-AI plots in Fig. 15, the Centre for Environmental Data Analysis (CEDA) and the Met Office for providing access
5    to the Met Office LIDARNET data, the Department for Transport and the Civil Aviation authority for funding the Met Office Raman Lidar network, EUMETSAT for providing the analysis of SEVIRI data, the NASA CALIOP and CATS teams for providing access to plots on the Web. Funding for Draude was provided by the U.K. Natural Environment Research Council through Manchester–Liverpool Doctoral Training Programme Grant NE/L002469/1. Partial funding for Vaughan and Schultz was provided by the Natural Environment Research Council to the University of Manchester through Grant NE/I005234/1.




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





**Figure 1.** Location of lidar stations used in this paper. Blue and green circles denote Raman lidars, red denote Lufft CHM 15k ceilometers, black denotes the two radiosonde stations mentioned in the text



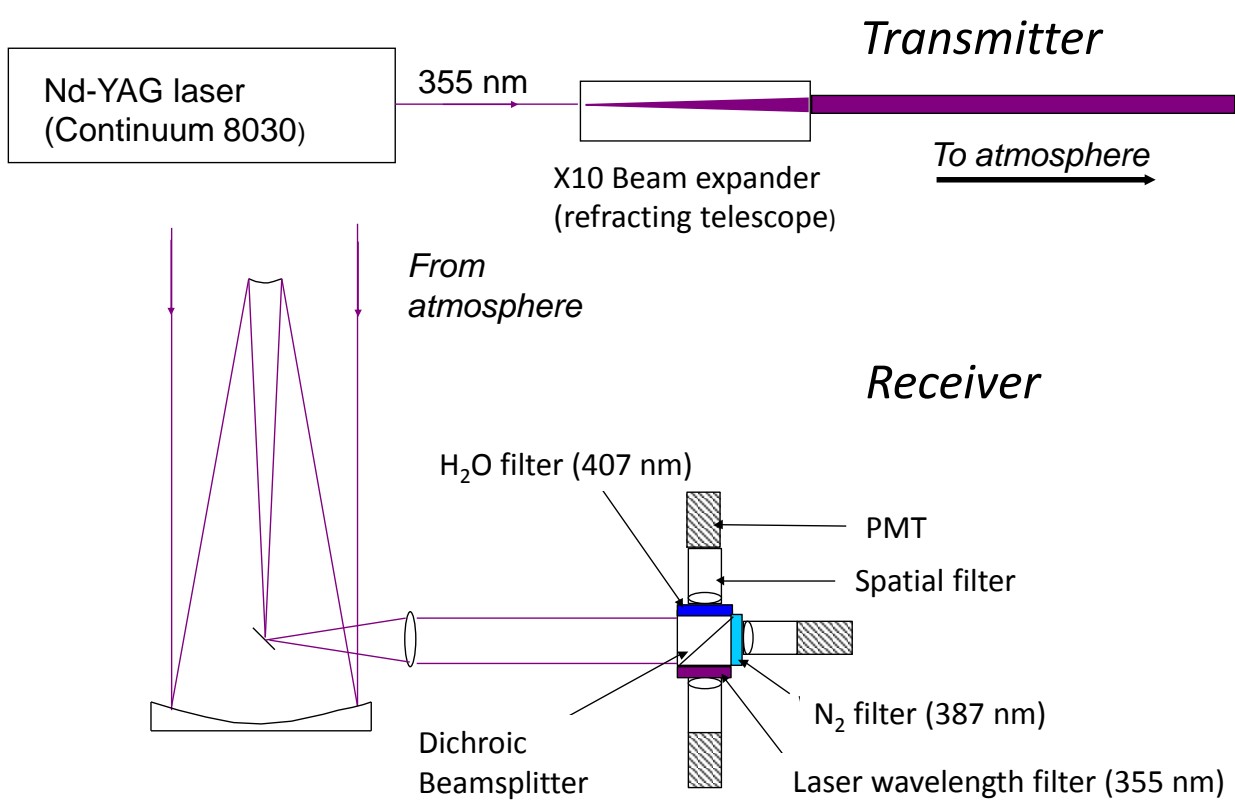

**Figure 2.** Schematic diagram of the Capel Dewi Raman lidar





**Figure 3.** Normalised ratios of elastic:Raman signals (black, normalised to 1) and Raman:synthetic molecular signals (blue, normalised to 1.5), using Capel Dewi lidar data from the night of 23–24 May 2016, between 2149 UTC and 0309 UTC. The Castor Bay radiosonde profile for 0000 UTC on 24 May 2016 was used for the molecular profile. Five distinct layers are identified, each by horizontal lines at their boundaries. The AOD is also shown for each layer.

**Figure 4.** As Fig. 3 but using the radiosonde from Camborne at 0000 UTC on May 24 2016



**Figure 5.** Variation of backscatter signal with time and height over the course of 23 May 2016 at three Met Office stations. The black marks indicate cloud bases. The dark blue and green layers above 2 km identified by the red arrows depict the aerosol layers.





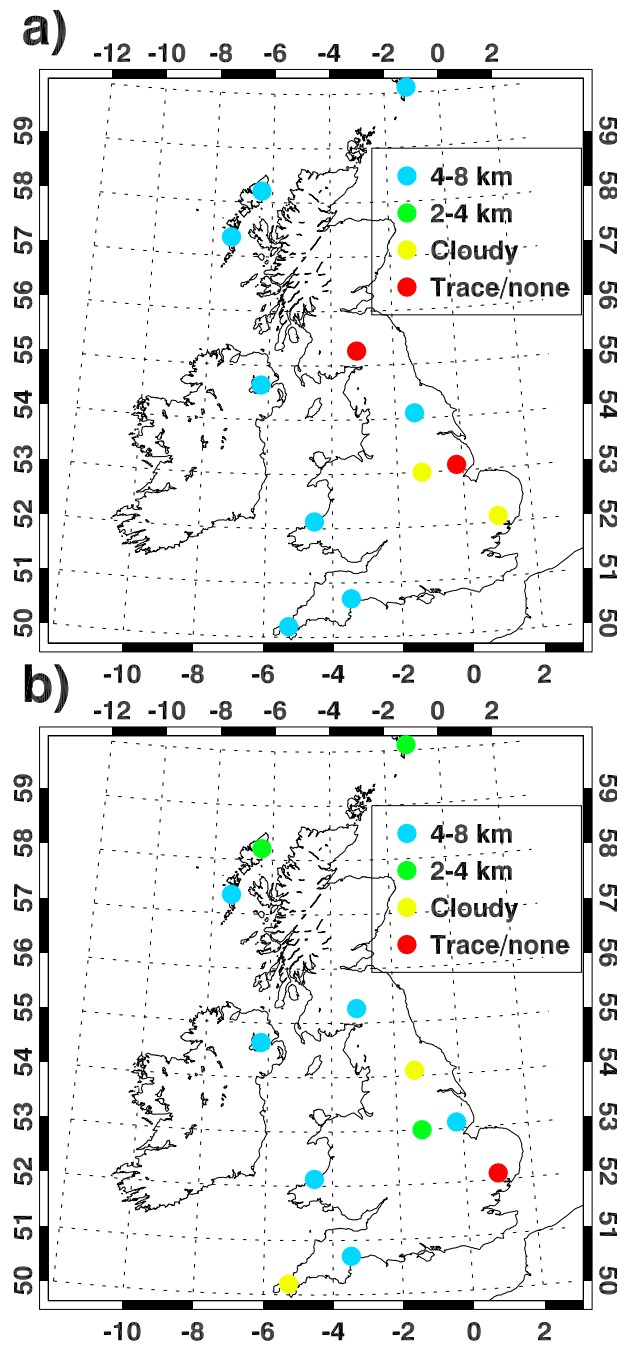

**Figure 6.** Aerosol observations by the Lufft CHM 15 k ceilometers. a) 23 May, b) 24 May.





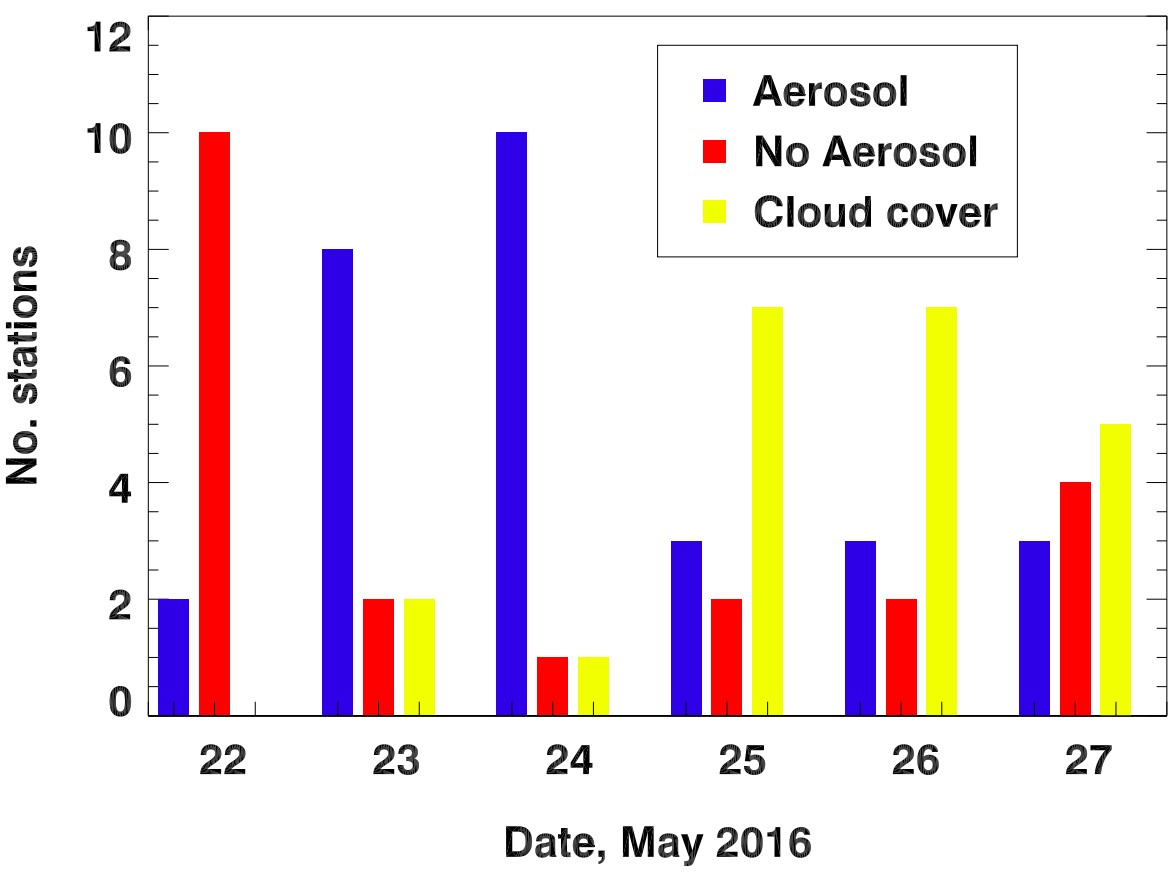

**Figure 7.** The number of Lufft ceilometers that observed distinct layers of aerosol (blue), trace amounts or no aerosol (red), or were obscured by clouds (yellow) for the period 22 to 27 May 2016





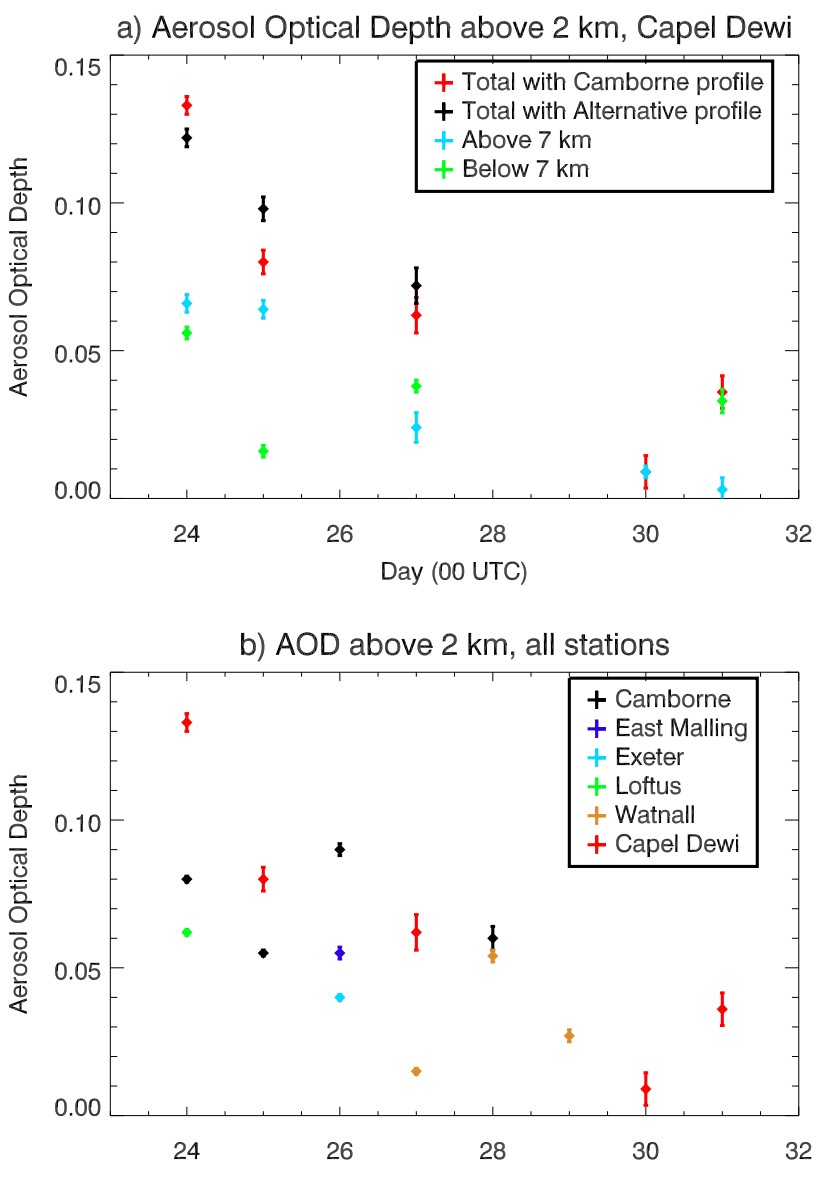

**Figure 8.** Total aerosol optical depth measured above 2 km by the Raman lidars during the last 8 nights of May 2016, expressed as whole-night averages. a) Capel Dewi measurements, analysed with two different radiosonde profiles, with a division to 2-7 km and >7 km using the Camborne radiosonde. Statistical error bars are shown on each point but the systematic error associated with the choice of radiosonde profile clearly dominates. b) Total AOD above 2 km for all the stations. The Capel Dewi points are those calculated using the Camborne radiosonde, as in a).





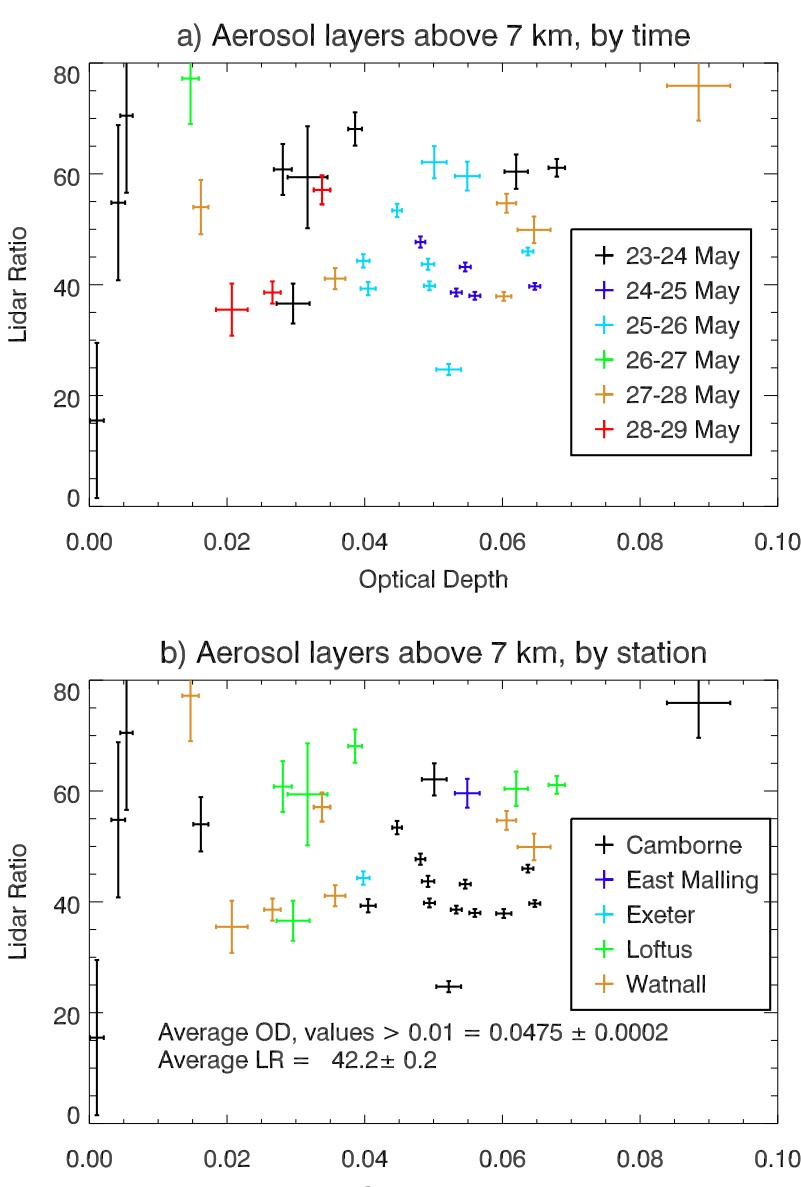

**Figure 9.** Lidar ratio plotted against total aerosol optical depth measured above 7 km by the Met Office Raman lidars a) according to time, b) according to station.





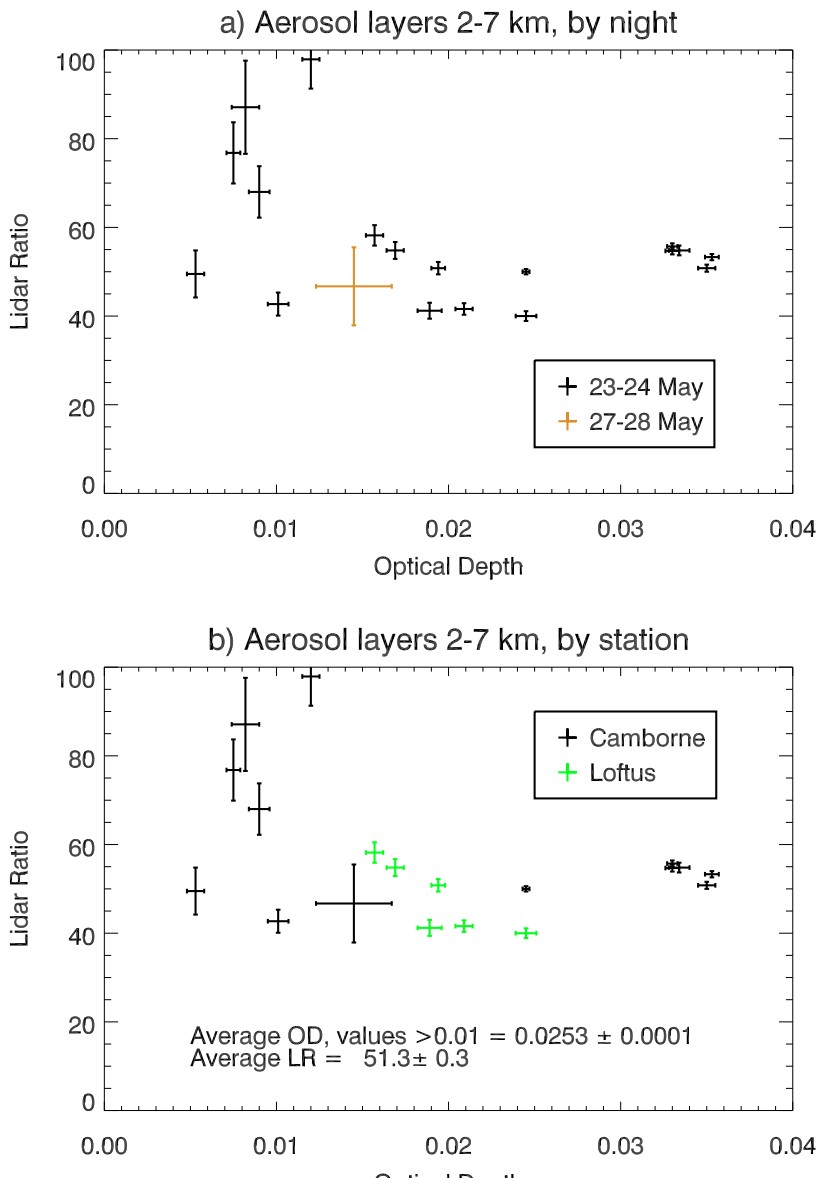

**Figure 10.** Lidar ratio plotted against total aerosol optical depth measured from 2 to 7 km by the Met Office Raman lidars a) according to time, b) according to station.





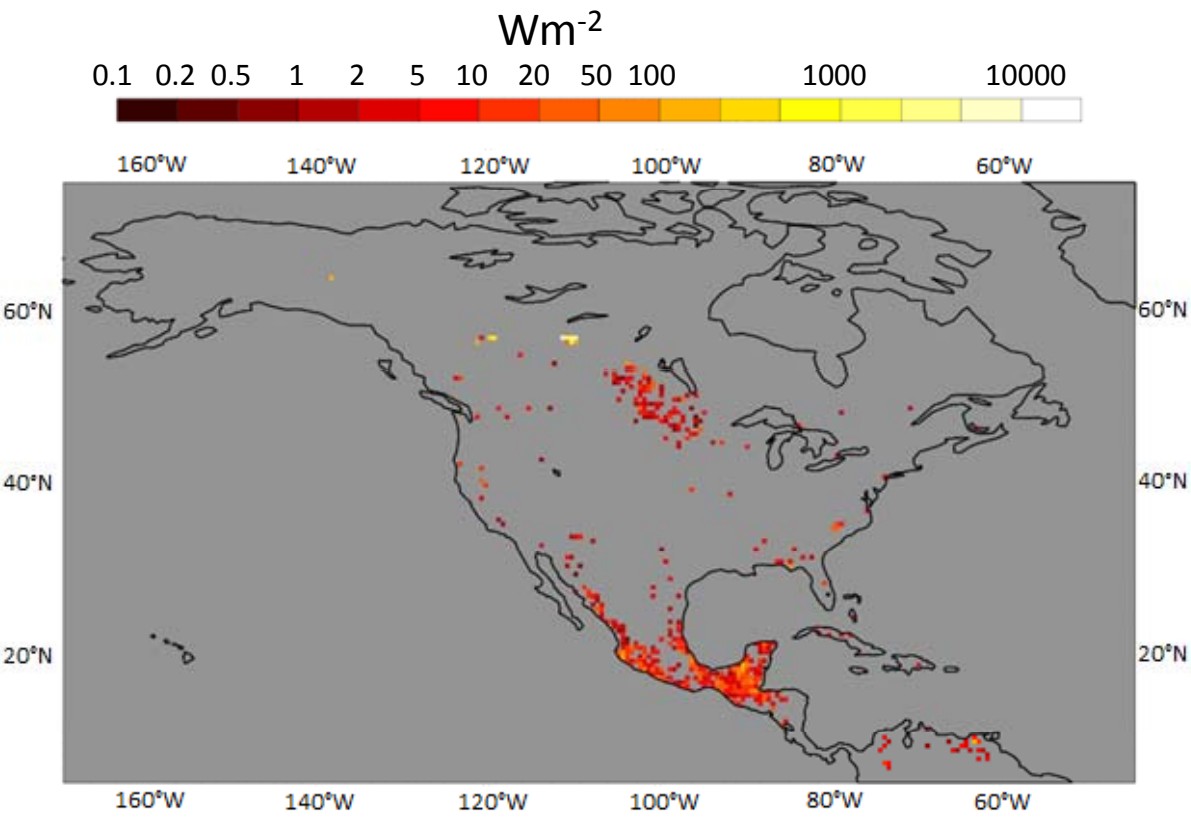

**Figure 11.** CAMS daily fire product for 17 May 2016. Colours show the average of the observed fire radiative power areal density, in W m$^{-2}$.





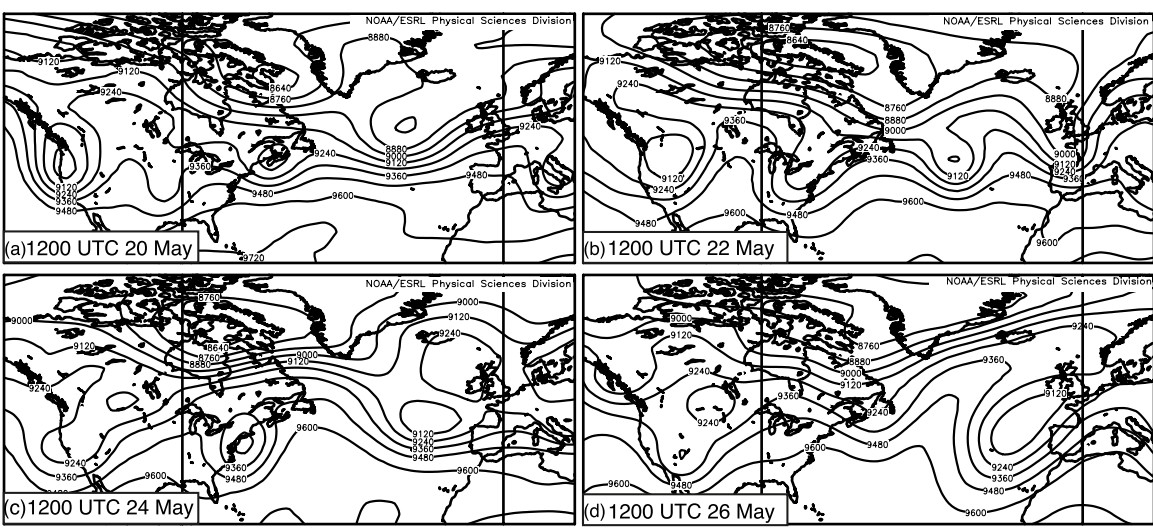

**Figure 12.** 300–hPa geopotential height (m) for 20–26 May 2016 from NCEP–NCAR reanalyses (Kalnay et al., 1996), showing how the zonal flow developed into a blocking pattern.



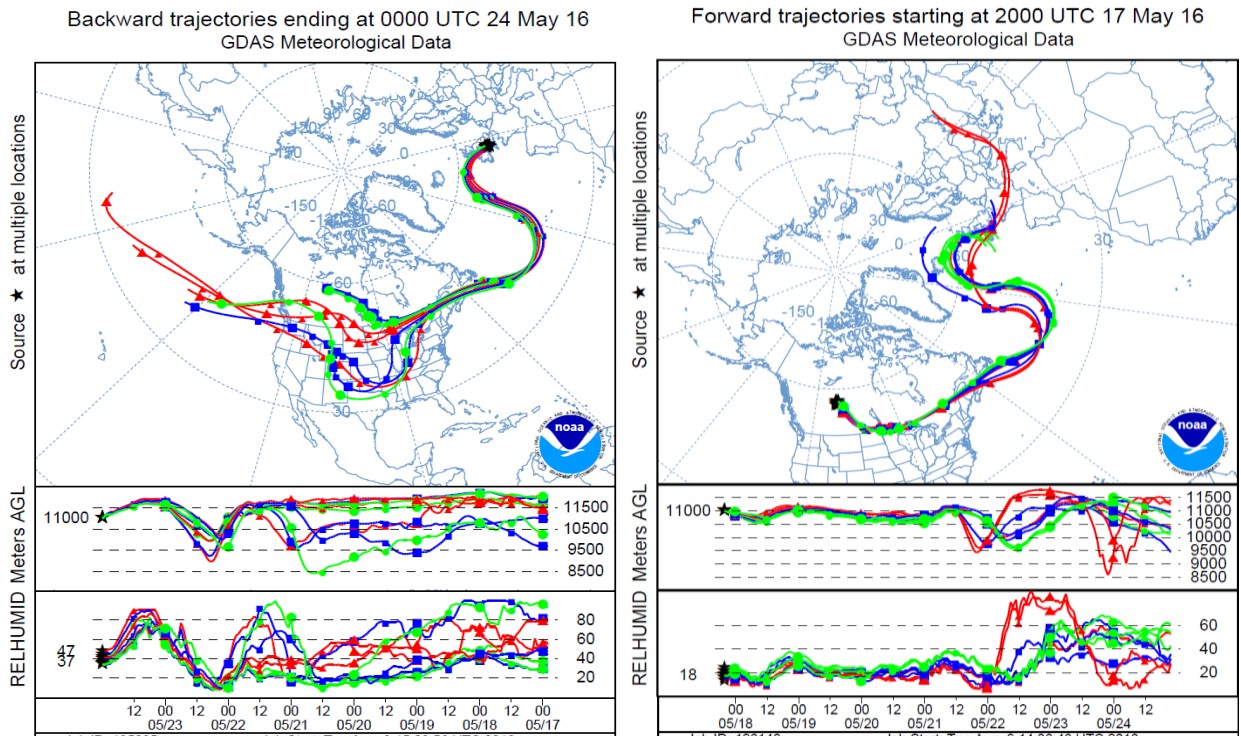

**Figure 13.** Back-trajectories calculated by HYSPLIT. Left: 9 back-trajectories started at 11 km from a square 1° wide over Capel Dewi at 0000 UTC 24 May 2016; Right: 9 forward-trajectories started at 11 km from a square 1° wide over 56.5°N, 111.5°W at 2000 UTC 17 May 2016





**Figure 14.** Dispersion of a column of particles initiated between 3 and 10 km over Fort McMurray at 2000 UTC on 17 May, calculated by the HYSPLIT on-line model for 0800 UTC on 21 May.



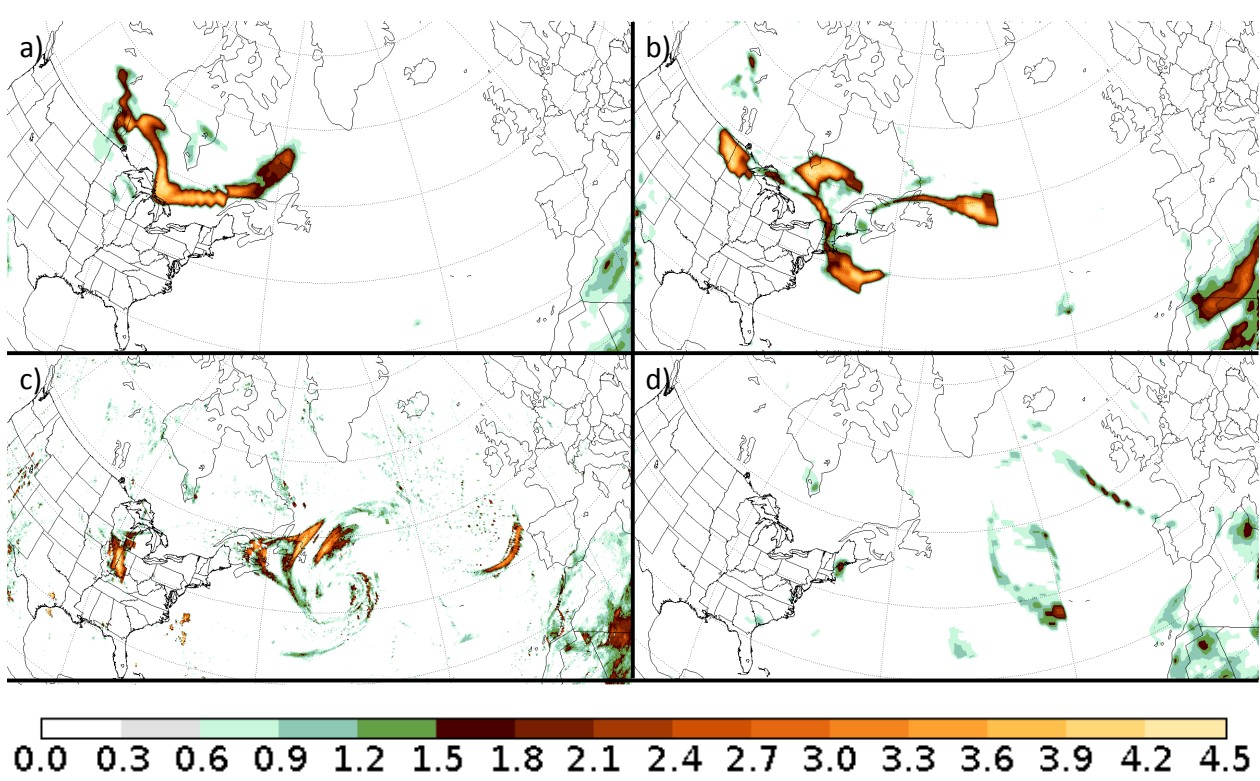

**Figure 15.** OMPS Aerosol Index for sections of the S-NPP orbits at 0500 UTC on a) 19, b) 20, c) 21 and d) 22 May. Images courtesy of Colin Seftor.





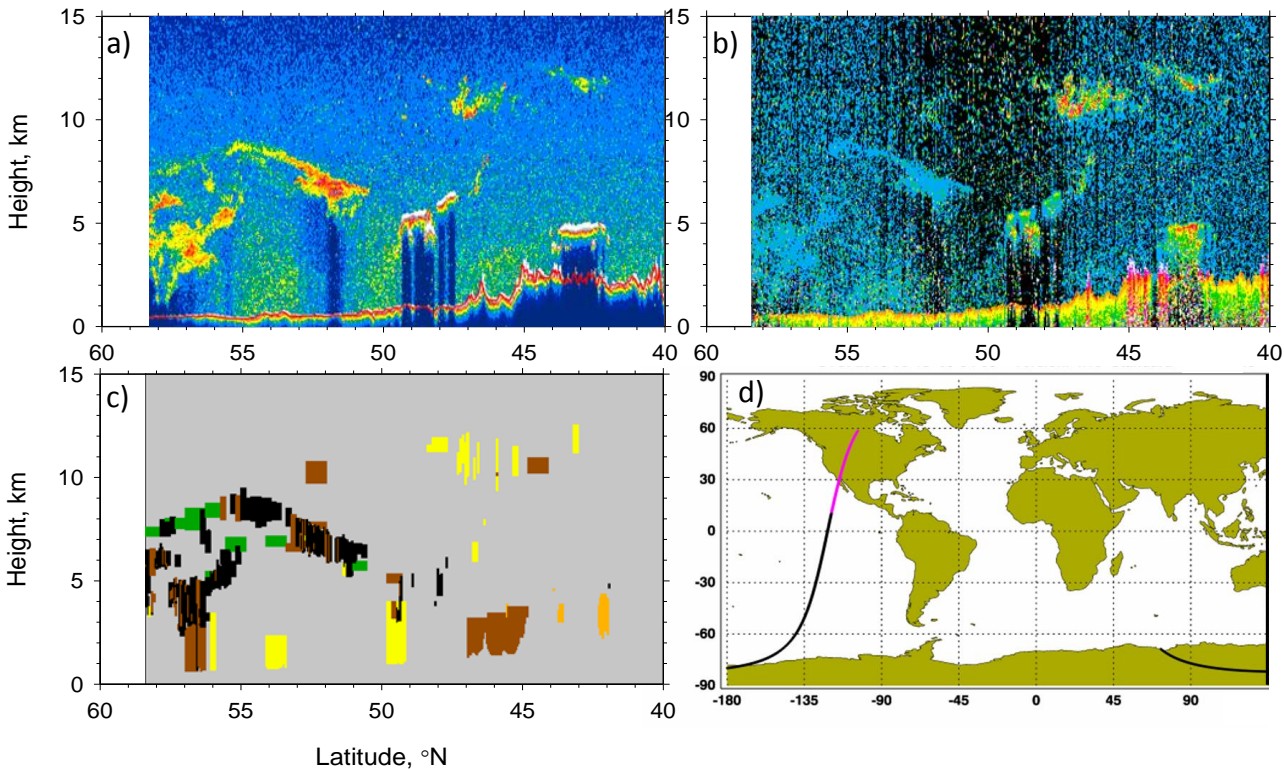

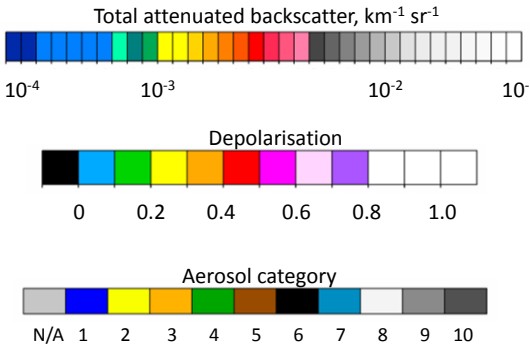

**Figure 16.** (a) Total attenuated backscatter at 532 nm, $\text{km}^{-1}\ \text{sr}^{-1}$; (b) volume depolarisation ratio; (c) aerosol subtype plotted against position for (d) a section of the CALIPSO orbit (shown in pink) from 0934 to 0939 UTC on 18 May 2016. Figure adapted from on-line figures on CALIPSO website. Colour bars for each panel have been expanded and are shown separately for clarity. Aerosol categories 5 (brown, polluted aerosol) and 6 (smoke) are of most interest to this paper.



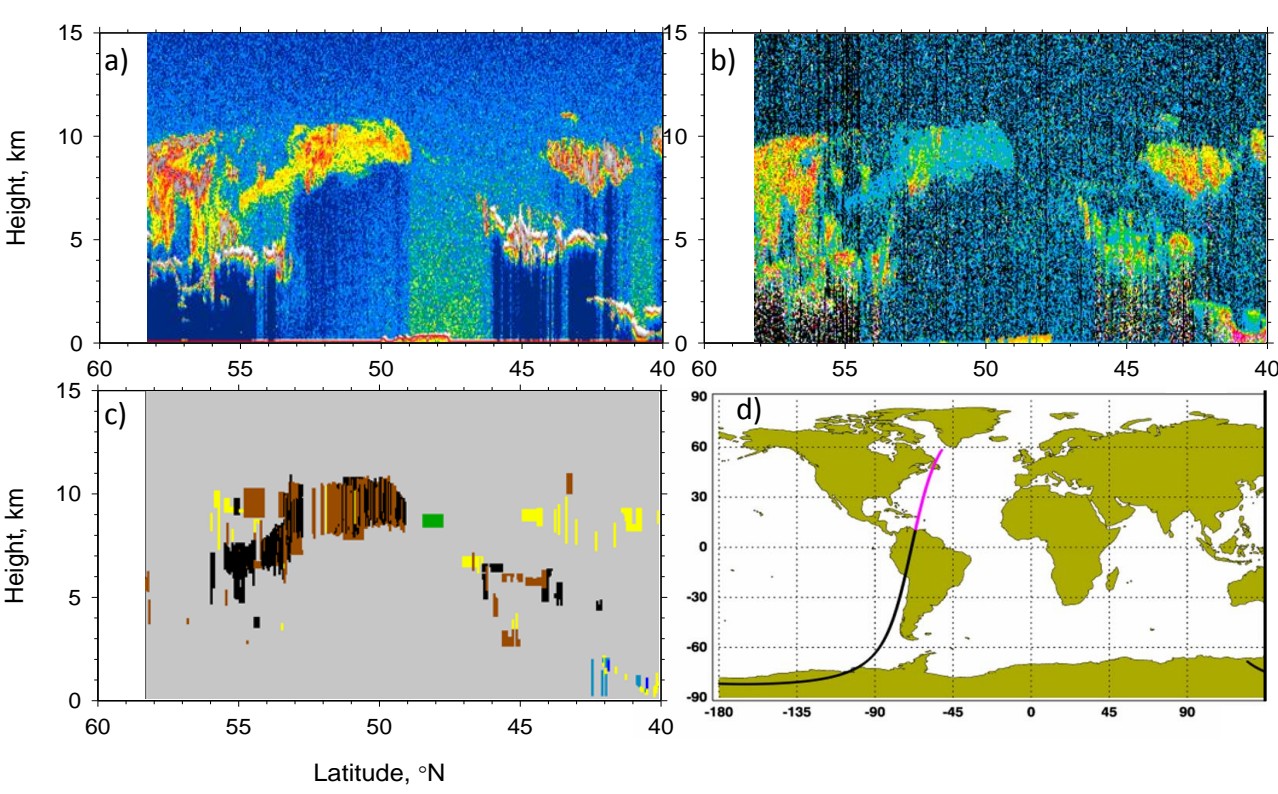

**Figure 17.** As Fig. 16 for a section of the CALIPSO orbit which passed over the Atlantic from 0604 to 0609 UTC on 20 May 2016.





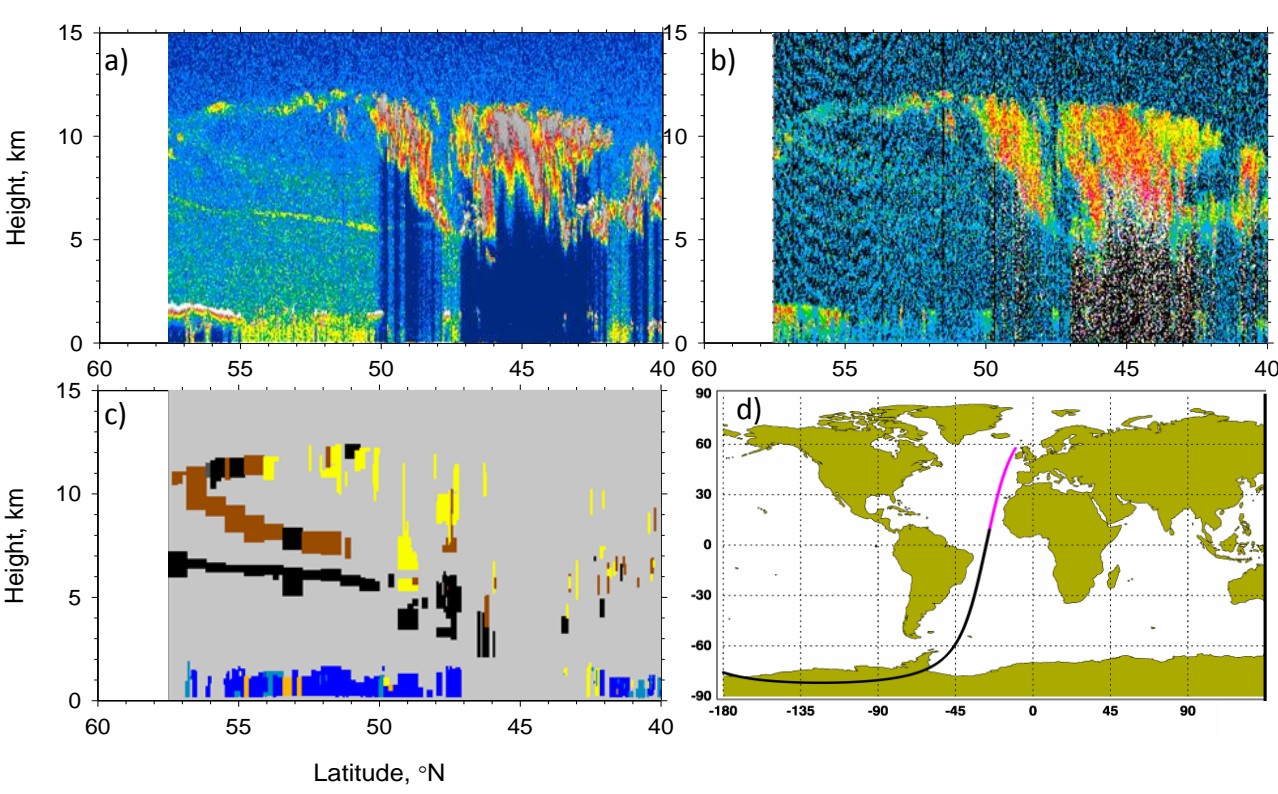

**Figure 18.** As Fig. 16 for a section of the CALIPSO orbit which passed over the Atlantic from 0317 to 0322 UTC on 23 May 2016.



**Figure 19.** The total attenuated backscatter (top) and perpendicularly polarized backscatter (middle) signals from the CATS lidar around 0216 on 22 May, adapted from figures on the CATS website. Colours are the same as CALIPSO (Fig. 16). Also shown is the path of the ISS and (in green) where the measurements were taken (bottom).





**Figure 20.** SEVIRI Day Natural Colour RGB images from 1830 UTC 22 May (top left); 0445 UTC 23 May, top right; 1945 UTC 23 May, bottom left; and 1930 UTC 24 May (bottom right). Smoke appears as faint blue-grey streaks. The red-brown streak in the North Sea shown in the bottom right image is the shadow of a smoke streak on some low-lying water clouds. Images taken from EUMETSAT web site.