# Peer review of "Transport of Canadian forest fire smoke over the UK as observed by lidar"

_Atmospheric Chemistry and Physics, 2017_

## Short Comment (SC1) · 18 Jan 2018

Probably that presented in the introduction the scientific background of the problems of the long-range transport of combustion products in the atmosphere would be more complete when mentioning also the cases of anomalous (eastern) long-range transport of smoke and CO from the forest fires in North America to the Northern Eurasia in August 2004 and that from the forest fires in Central Siberia to Central Europe in July 2016, associated with atmospheric blocking of the dipole type (doi: 10.3103/S1068373917080052 and doi: 10.1134/S1028334X17070261).

---

## Referee Comment (RC1) · Anonymous Referee #1 · 16 Feb 2018

General: The paper presents interesting observations of Canadian fire smoke over UK and is appropriate for APC. It brings together different observations of ground-based and spaceborne lidars on smoke layers. However, the paper is a bit lengthy. There is a good chance to make it more compact (see my detailed comments) and thus more interesting for a broader aerosol science community. Minor revisions are required.

Details:

Abstract: How much is 'weak depolarization'? Please provide numbers . . . . <5%... ?

Introduction: The paper of . . .. Alados-Arboledas, L., D. Müller, J. L. Guerrero-Rascado, F. Navas-Guzmán, D. Pérez-Ramírez, and F. J. Olmo (2011), Optical and microphysical properties of fresh biomass burning aerosol retrieved by Raman lidar, and star-and

sun-photometry, Geophys. Res. Lett., 38, L01807, doi:10.1029/2010GL045999. . ..
should be included in the references. . .

Wandinger (JGR, 2002) was probably one the first who analyzed Canadian smoke lidar
data (measured in 1998), and Mattis (GRL 2003, JGR 2008) also from the Leipzig lidar
group studied many smoke layers from North America, and Murayama (GRL, 2004)
made Raman lidar observations in Siberian smoke. . .

2. Instrumentation: The description of the Capel Dewi Raman lidar is very long. This
is a lidar application paper so that so many lidar instrumental details are not needed. It
is sufficient to mention the measurement channels and the products you can derive.

The same is true for the Raymetrics lidar systems, too much technical information
which is not needed.

3 Retrieval. . .

Basic lidar equations (2) and (3) are not needed!

Figure 4 is not needed. Figure 3 is fine, to give an example of basic profiles of lidar
products. Figure 4 is not needed, but triggers the question: Why do you not use
just modeled, ECMWF or GDAS, temperature and pressure profiles in the lidar data
analysis. These profiles are usually more appropriate than radiosonde data because
the model data consider all the available radiosonde information and are available at
model grid points close to the lidar site and for the given lidar measurement period.

4 Results. . .

Figure 5 is ok.

Figure 6: It is sufficient to show the 23 May case only.

Figure 7: I would remove this figure! At least, I do not need it to understand the paper
and to get the main message of the paper.

Figure 8: I would show Figure 8b only, and symbols should be larger and different (circles, squares, triangles. . .), please use more contrast rich colors, orange, blue, green, red.

Figure 9: I would show Figure 9b only. Again use large and different symbols and contrasting colors.

Figure 10b is sufficient, same comments regarding colors and symbols as above.

Concerning depolarization ratio at 355 nm, and the potential interpretation with respect to smoke, please check the Burton et al. paper (ACP 2015 paper, triple wavelength depol ratio ). They measured a Canadian smoke layer with 355 nm particle depol ratio of 21% whereas they found only 15% at 532 nm and less than 2% at 1064 nm.

Burton, S. P., Hair, J. W., Kahnert, M., Ferrare, R. A., Hostetler, C. A., Cook, A. L., Harper, D. B., Berkoff, T. A., Seaman, S. T., Collins, J. E., Fenn, M. A., and Rogers, R. R.: Observations of the spectral dependence of linear particle depolarization ratio of aerosols using NASA Langley airborne High Spectral Resolution Lidar, Atmos. Chem. Phys., 15, 13453-13473, https://doi.org/10.5194/acp-15-13453-2015, 2015.

5 Origin of aerosols

The discussion is very long, can be shortened easily. Please focus on the main messages.

Is Figure 12 needed? We have trajectories in Figure 13 and all the convincing space-borne observations in Figs. 16, 17, 18, and 19!

Figure 14! To my opinion, the figure is not needed.

This is to my opinion a lidar paper, so I would skip Figure 15 and all the lengthy explanations of AI.

Figure 16 is nice. Please save space by a compact and optimized arrangement of the color scales.

Figure 17 is fine as well. There is a thick (attenuating) smoke layer and low depolarization ratio. Please be very accurate in the description: CALIOP is providing volume linear depolarization ratios, please state that always clearly, and this quantity can vary strongly because of the changing total/Rayleigh backscatter ratio. . ... if we would have the particle linear depolarization ratio (instead of the volume depolarization ratio) then we would probably have always the same values. . .. But it is clear, and this an interesting aspect, the smoke particle depolarization ratio is significantly smaller than the one for cirrus. Thus, the particle depolarization ratio can be nicely used to distinguish between cirrus and smoke at, e.g., 10 km height were both can be present at the same time. . ..

Figure 20. . .! I do not see much. I would remove this figure.

At the end, the ground-based and spaceborne lidar observations of smoke are the highlight of the paper. And these measurements should be presented in a more condensed form. That would improve the paper. However, feel free to accept or reject my comments to the figures.

---

## Referee Comment (RC2) · Anonymous Referee #2 · 23 Mar 2018

This paper describes lidar observations of a Canadian forest fire smoke event over UK. It provides technical details of various types of ground-based lidars, and combines space-born observations and back-trajectory model to trace the origin of the smoke layers observed by the lidars in UK to Canadian fires. The reviewer thinks the technical description and result sections can be better balanced in two potential ways with different focuses. One way is to keep the detailed description of different instruments and add more on uncertainty estimates (something like Fig.3,4 where different atmospheric background profiles are taken, and that resulted in different layered AOD estimates. And what if modeled atmospheric background profile is taken?), and give an overview of strength and weakness of different lidars on observations of thin layers of aerosols. This would make the paper more appropriate for Atmospheric

[Figure]

Measurement Techniques. The other way is to dig deeper on this one case study on not only the observations of smoke layers, but also the cause of multi layers and long lingering time (through analysis of the evolving meteorology, and the evolution of the smoke over the Atlantic using space-borne observations), which will really make this study interesting. This would help readers have a whole picture of this one special case. The second way, with a focus on the observation and mechanism of this special smoke event, appears more interesting to the reviewer. So the comments below follow that line. 1. Technical descriptions of the lidars are lengthy. Some basic equations and tables/figures (eg., equations 2, 3, figure 2) can be cut and text be shortened to 1/2 $\sim$2/3. 2. The authors provide multiple ways to show that the origin of the smoke is Canada. However it is lengthy. For example, Fig 13, 14 both present HYSPLIT results, "However examples like this proved rare" as admitted by the authors. So one of the figures is sufficient. Also the SEVIRI image can be provided as supplemental material. And the CALIPSO figures can be condensed into one or two. 3. The Canadian smoke evolved with a mid-latitude cyclone system during its transport over the Atlantic before it reached UK. This makes it a very interesting and special case. See the true-color imaginary from Terra for May 22, 2016 (https://worldview.earthdata.nasa.gov/?p=geographic&l=VIIRS_SNPP_CorrectedReflectance_TrueColor(hidden),MODIS_ 05-28&z=3&v=-144.34275416756438,-5.132810354232753,58.43849583243563,106.94531464576724).
It is worth discussing how and why the smoke evolved from one big blob (CALIOP observations) to multiple layers. The authors described a little bit about the upper level (300hPa) patterns, but it would also be desired to include some vertical cross section analysis for atmospheric states, including wind and moisture (which impacts smoke depolarization), and their evolution. 4. Page 7, line 4, It would be helpful to include typical values of delta_a for fresh smoke and aged smoke (from other studies), just for informational and comparison purpose for readers.

[Figure]

**Fig. 1.**

---

## Author Comment (AC1) · 5 Jun 2018

Reference to smoke from Siberia now included.

---

## Author Comment (AC2) · 5 Jun 2018

**Response to Reviewer 1**

*General: The paper presents interesting observations of Canadian fire smoke over UK and is appropriate for APC. It brings together different observations of ground-based and spaceborne lidars on smoke layers. However, the paper is a bit lengthy. There is a good chance to make it more compact (see my detailed comments) and thus more interesting for a broader aerosol science community. Minor revisions are required.*
We thank the referee for the positive comments on the paper. We note the comment at the end of the review that the comments on the figures are advisory only.

Details:
*Abstract: How much is 'weak depolarization'? Please provide numbers . . .. <5%... ?*
Now provided.

*Introduction: The paper of . . .. Alados-Arboledas, L., D. Müller, J. L. Guerrero-Rascado, F. Navas-Guzmán, D. Pérez-Ramírez, and F. J. Olmo (2011), Optical and microphysical properties of fresh biomass burning aerosol retrieved by Raman lidar, and star-and sun-photometry, Geophys. Res. Lett., 38, L01807, doi:10.1029/2010GL045999. . .. should be included in the references. . .*
*Wandinger (JGR, 2002) was probably one the first who analyzed Canadian smoke lidar data (measured in 1998), and Mattis (GRL 2003, JGR 2008) also from the Leipzig lidar group studied many smoke layers from North America, and Murayama (GRL, 2004) made Raman lidar observations in Siberian smoke. . .*
These references are now included in the paper

*2. Instrumentation: The description of the Capel Dewi Raman lidar is very long. This is a lidar application paper so that so many lidar instrumental details are not needed. It is sufficient to mention the measurement channels and the products you can derive. The same is true for the Raymetrics lidar systems, too much technical information which is not needed.*
We have moved the instrumental details to a supplement. These instruments have not been described previously in the literature and therefore we feel they should be available should people want to find them.

*3 Retrieval. . .*
*Basic lidar equations (2) and (3) are not needed!*
This section also moved to the supplement

*Figure 4 is not needed. Figure 3 is fine, to give an example of basic profiles of lidar products. Figure 4 is not needed, but triggers the question: Why do you not use just modeled, ECMWF or GDAS, temperature and pressure profiles in the lidar data analysis. These profiles are usually more appropriate than radiosonde data because the model data consider all the available radiosonde information and are available at model grid points close to the lidar site and for the given lidar measurement period.*
We disagree: fig.4 provides an estimate of the kind of systematic error introduced by the choice of temperature profile. The same kind of error would arise if we used a model profile, and could be evaluated in a similar way, but we do not agree that a model profile is necessarily better than a measured one, even if it is supposed to be at the same place and time. Our choice of measured profiles was made because they are readily available, and capture the height of the tropopause more precisely than a model profile. As we are interested in aerosols near the tropopause, a systematic error in the height of the tropopause is something we wanted to minimise.

*4 Results. . .*
*Figure 5 is ok.*
*Figure 6: It is sufficient to show the 23 May case only.*
We have kept 24 May as well as the two figures together show how the 4-8 km aerosol spread SE across the UK, and 2-4 km aerosol began to appear.

*Figure 7: I would remove this figure! At least, I do not need it to understand the paper and to get the main message of the paper.*
Again, we prefer to keep this figure as it summarised the ceilometer data

*Figure 8: I would show Figure 8b only, and symbols should be larger and different (circles, squares, triangles. . .), please use more contrast rich colors, orange, blue, green, red.*
*Figure 9: I would show Figure 9b only. Again use large and different symbols and contrasting colors.*
*Figure 10b is sufficient, same comments regarding colors and symbols as above.*
We have reduced these to one panel each and changed the symbols.

*Concerning depolarization ratio at 355 nm, and the potential interpretation with respect to smoke, please check the Burton et al. paper (ACP 2015 paper, triple wavelength depol ratio ). They measured a Canadian smoke layer with 355 nm particle depol ratio of 21% whereas they found only 15% at 532 nm and less than 2% at 1064 nm. Burton, S. P., Hair, J. W., Kahnert, M., Ferrare, R. A., Hostetler, C. A., Cook, A. L., Harper, D. B., Berkoff, T. A., Seaman, S. T., Collins, J. E., Fenn, M. A., and Rogers, R. R.: Observations of the spectral dependence of linear particle depolarization ratio of aerosols using NASA Langley airborne High Spectral Resolution Lidar, Atmos. Chem. Phys., 15, 13453-13473, https://doi.org/10.5194/acp-15-13453-2015, 2015.*
Thanks, we have included a discussion of this paper

*5 Origin of aerosols*
*The discussion is very long, can be shortened easily. Please focus on the main messages.*
*Is Figure 12 needed? We have trajectories in Figure 13 and all the convincing spaceborne observations in Figs. 16, 17, 18, and 19!*
We need this figure to show what we mean by an atmospheric block

*Figure 14! To my opinion, the figure is not needed.*
OK, we have removed this

*This is to my opinion a lidar paper, so I would skip Figure 15 and all the lengthy explanations of AI.*
No, it isn't just a lidar paper, and we are trying to use all the information we have on the spread of the aerosol. We can't rely on trajectories, so we have to use observations to follow the smoke plume across the Atlantic. Therefore we have to describe these observations so that the reader understands what they mean. See also the opinion of referee 2, who wanted more discussion of the meteorology.

*Figure 16 is nice. Please save space by a compact and optimized arrangement of the color scales.*
Done

*Figure 17 is fine as well. There is a thick (attenuating) smoke layer and low depolarization ratio. Please be very accurate in the description: CALIOP is providing volume linear depolarization ratios, please state that always clearly, and this quantity can vary strongly because of the changing total/Rayleigh backscatter ratio. . ... if we would have the particle linear depolarization ratio (instead of the volume depolarization ratio) then we would probably have always the same values. . .. But it is clear, and this an interesting aspect, the smoke particle depolarization ratio is significantly smaller than the one for cirrus. Thus, the particle depolarization ratio can be nicely used to distinguish between cirrus and smoke at, e.g., 10 km height were both can be present at the same time. . ..*
We now emphasise that it's the volume depolarisation ratio.

*Figure 20. . .! I do not see much. I would remove this figure.*
This is now moved to the Supplementary material

---

## Author Comment (AC3) · 5 Jun 2018

**Response to Reviewer 2**

*This paper describes lidar observations of a Canadian forest fire smoke event over UK. It provides technical details of various types of ground-based lidars, and combines space-born observations and back-trajectory model to trace the origin of the smoke layers observed by the lidars in UK to Canadian fires. The reviewer thinks the technical description and result sections can be better balanced in two potential ways with different focuses. One way is to keep the detailed description of different instruments and add more on uncertainty estimates (something like Fig.3,4 where different atmospheric background profiles are taken, and that resulted in different layered AOD estimates. And what if modeled atmospheric background profile is taken?), and give an overview of strength and weakness of different lidars on observations of thin layers of aerosols. This would make the paper more appropriate for Atmospheric Measurement Techniques.*
The instrument details have now been moved to the Supplement. See also response to Referee 1

*The other way is to dig deeper on this one case study on not only the observations of smoke layers, but also the cause of multi layers and long lingering time (through analysis of the evolving meteorology, and the evolution of the smoke over the Atlantic using space-borne observations), which will really make this study interesting. This would help readers have a whole picture of this one special case. The second way, with a focus on the observation and mechanism of this special smoke event, appears more interesting to the reviewer. So the comments below follow that line.*

*1. Technical descriptions of the lidars are lengthy. Some basic equations and tables/figures (eg., equations 2, 3, figure 2) can be cut and text be shortened to 1/2 _2/3.*
This text now moved to the supplement

*2. The authors provide multiple ways to show that the origin of the smoke is Canada. However it is lengthy. For example, Fig 13, 14both present HYSPLIT results, "However examples like this proved rare" as admittedby the authors. So one of the figures is sufficient.*
Agreed, fig 14 removed.

*Also the SEVIRI image can be provided as supplemental material. And the CALIPSO figures can be condensed into one or two.*
SEVIRI image moved to supplement. Guided by referee 1'c comments we have not reduced the CALIPSO figures.

*3. The Canadian smoke evolved with a mid-latitude cyclone system during its transport over the Atlantic before it reached UK. This makes it a very interesting and special case. See the true-color imaginary from Terra for May 22, 2016 (https://worldview.earthdata.nasa.gov/?p=geographic&l=VIIRS_SNPP_CorrectedReflectance_TrueColor(hidden),MODIS_05-28&z=3&v=-144.34275416756438,-5.132810354232753,58.43849583243563,106.94531464576724). It is worth discussing how and why the smoke evolved from one big blob (CALIOP observations) to multiple layers. The authors described a little bit about the upper level (300hPa) patterns, but it would also be desired to include some vertical cross section analysis for atmospheric states, including wind and moisture (which impacts smoke depolarization), and their evolution.*
We do not wish to expand the number of figures in this paper or to discuss in detail the processes responsible for generating the layered structure, but we have added text on pp 6, 9 and 10 drawing attention to this evolution. In fact, the OMPS figure shows very clearly how the aerosol became entrained into a cyclone, and we use this as the basis for discussion. We also refer to Dacre and Harvey's recent paper on the dispersion of atmospheric trajectories in blocking situations.

*4. Page 7, line 4, It would be helpful to include typical values of delta_a for fresh smoke and aged smoke (from other studies), just for informational and comparison purpose for readers.*

We now include such comparisons on p.6. Unfortunately most of the literature is for 532 and 1064 nm and there are few published measurements at 355 nm.

---

## Author Response (AR2)

Author's response

Sentences added at the end of the abstract and conclusions as requested by the Editor.